# International Approaches to Management of CFTR-Related Metabolic Syndrome/Cystic Fibrosis Screen Positive, Inconclusive Diagnosis

**DOI:** 10.3390/ijns8010005

**Published:** 2022-01-11

**Authors:** Jane Chudleigh, Jürg Barben, Clement L. Ren, Kevin W. Southern

**Affiliations:** 1School of Health Sciences, City, University of London, London EC1V 0HB, UK; 2Children’s Hospital of Eastern Switzerland, 9000 St. Gallen, Switzerland; Juerg.Barben@kispisg.ch; 3Children’s Hospital of Philadelphia, Philadelphia, PA 19104, USA; renc@chop.edu; 4Department of Women’s and Children’s Health, University of Liverpool, Liverpool L69 3BX, UK; kwsouth@liv.ac.uk

**Keywords:** cystic fibrosis, newborn bloodspot screening, CFTR-related metabolic syndrome, cystic fibrosis screen positive, inconclusive diagnosis

## Abstract

The main aim of the present study was to explore health professionals’ reported experiences and approaches to managing children who receive a designation of cystic fibrosis transmembrane conductance regulator-related metabolic syndrome/cystic fibrosis screen positive inconclusive diagnosis following a positive NBS result for cystic fibrosis. An online questionnaire was distributed via Qualtrics Survey Software and circulated to a purposive, international sample of health professionals involved in managing children with this designation. In total, 101 clinicians completed the online survey: 39 from the US, six from Canada, and 56 from Europe (including the UK). Results indicated that while respondents reported minor deviations in practice, they were cognizant of recommendations in the updated guidance and for the most part, attempted to implement these into practice consistently internationally. Where variation was reported, the purpose of this appeared to be to enable clinicians to respond to either clinical assessments or parental anxiety in order to improve outcomes for the child and family. Further research is needed to determine if these findings are reflective of both a wider audience of clinicians and actual (rather than reported) practice.

## 1. Introduction

Newborn screening (NBS) for cystic fibrosis (CF) has many benefits, including improved health outcomes for the affected child [1,2]. Screening algorithms for CF differ internationally [3,4]. First-tier testing generally consists of measuring immunoreactive trypsinogen (IRT); second-tier testing differs considerably between programmes and includes IRT, pancreatitis-associated protein (PAP), and/or DNA. In most cases, diagnosis of CF after a positive NBS result is straightforward; the most reliable and widely available test for diagnosing CF is the sweat chloride (SC) test [5]. Following SC testing, most children with a positive NBS result will be confirmed as affected by or carriers of CF. However, identification of infants with an inconclusive diagnosis after a positive NBS result—termed cystic fibrosis transmembrane conductance regulator (CFTR)-related metabolic syndrome or CF Screen Positive Inconclusive Diagnosis (CRMS/CFSPID) is an increasingly recognized outcome [6]. Children with CRMS/CFSPID have either a normal sweat chloride (<30 mmol/L) and two CFTR mutations (at least one of which has unclear phenotypic consequences) or an intermediate sweat chloride value (30–59 mmol/L) and one or no CFTR mutations [7]. The incidence of CRMS/CFSPID varies internationally depending on the population and algorithms used; it is unclear how many of these children will go on to develop CF [8,9,10,11,12]. Whilst NBS protocols that do not employ DNA analysis will recognize significantly fewer infants with CRMS/CFSPID, this is often at the expense of performance with respect to positive predictive value and sensitivity [13].

Historically, internationally, there have been variations in terms of the way these children were managed and even the terminology used. Management of these infants is still evolving with increasing experience and reporting of relevant outcomes [13]. These children have, in most cases, a good prognosis and there is no evidence for improvement through early treatment. Those who convert to a CF diagnosis may benefit from early interventions to prevent long-term complications [13]. However, it is not clear which children this applies to or how frequently they should be monitored [10,11,13]; the reported proportion of children who convert from CRMS/CFSPID to a CF diagnosis and the age at which this occurs varies from 2–48% at <1–5 years of age [8,9,10,11,12,14]. Findings of studies suggest initial IRT and SC values are higher in children with a CF diagnosis following NBS compared to children with CRMS/CFSPID [9,10,11,12,14,15]. Additionally, *P. aeruginosa* colonisation was less common in children with CRMS/CFSPID compared to children with a diagnosis of CF [9,12,15]. The risk of developing a CFTR-related disorder defined as, “*…clinical conditions that are recognised to be associated with abnormality of the CFTR gene but are not CF*” [13], is not yet quantified. A limited number of studies have been conducted to explore how these children are being managed in practice. One study in Italy found varied practice with regard to sweat testing, chest x-ray, and salt supplementation [14]. Similarly, a study conducted in Switzerland also found varied practice with regard to sweat testing; only 16 children (53%) had a second sweat test and ongoing care; only half of the children with a CRMS/CFSPID designation were being cared for by a primary care physician (PCP)/general practitioner (GP) or a paediatrician [16].

In 2015, recommendations were made regarding the management of these infants, which included 31 consensus statements [7]. Updated guidance for the management of children with CRMS/CFSPID was published in 2020 [13]. These highlight lack of data currently available regarding the risk of infants with CRMS/CFSPID converting to a CF diagnosis. The nature of CRMS/CFSPID means it is difficult to provide clear and accurate information regarding long-term outcomes for these children. This uncertainty can cause parents additional anxiety [17,18,19], and therefore communication must be managed effectively but also consistently.

The main aim of the present study was to explore health professionals’ reported experiences and approaches to managing children who receive a designation of CRMS/CFSPID following a positive NBS result.

## 2. Materials and Methods

A questionnaire (Appendix A) was developed via Qualtrics Survey Software. This was circulated to a purposeful, international sample of health professionals involved in managing children with a CRMS/CFSPID designation. Ethical approval was granted by the Research Ethics Committee at City, University of London (ETH1920-0952).

The survey link was circulated to members of the European CF Society (ECFS) Neonatal Screening Working Group (NSWG) (*n* = circa 400 Worldwide, it is not known how many of these met the eligibility criteria) and representatives from each US state to the CF Foundation NBS quality improvement consortium (*n* = 50) between November 2020 and March 2021.

The online survey started with questions aimed at gathering demographic data such as country of work, job title, the number of years working with children with CF, and a CRMS/CFSPID designation. This was followed by questions related to clinicians’ reported experiences of designating the child as having CRMS/CFSPID, ongoing management of children with a CRMS/CFSPID designation, and their family, including if and how the consensus statements for CRMS/CFSPID are implemented in practice. Finally, communicating with professionals outside of the CF team about the child’s CRMS/CFSPID designation.

### Data Analysis

Quantitative (demographic and closed-ended questions) data were analysed using simple descriptive statistics. Qualitative data (open-ended questions) were analysed using qualitative content analysis [20]. An inductive approach was used focussing on the manifest meanings.

## 3. Results

### 3.1. Demographic Data

In total, 101 clinicians completed the online survey: 39 from the US, six from Canada, and 56 from Europe (including the UK). Participant characteristics including job title, years working with children with CF, and years working with children with CRMS/CFSPID can be seen in Table 1 and Table 2, respectively. The majority of respondents were doctors (*n* = 76, 75%), most had worked with children with CF for over 15 years (*n* = 73, 72%), but CRMS/CFSPID less than 15 years (*n* = 75, 74%).

### 3.2. Initial Consultation after a Positive NSB Result

Most respondents reported performing a multitude of tests during the initial assessment following the positive NBS result to reach the CRMS/CFSPID designation. These included: sweat test (*n* = 99, 98%); extended CFTR analysis (if the genotype was incomplete) (*n* = 82, 81%); collection of a stool sample for measurement of fecal elastase (*n* = 80, 79%) and less frequently: upper airway respiratory culture (e.g., oropharyngeal, nasopharyngeal or cough swab) (*n* = 13, 13%); chest x-ray (*n* = 4, 4%) and liver function tests and/or electrolytes (*n* = 2, 2%).

Following the initial assessment, in terms of information provision, respondents reported commonly discussing the fact that while the NBS result had been positive, further testing had indicated the result was inconclusive for CF:

“*I tell them that the newborn screen was abnormal but that the sweat test was not positive for cystic fibrosis*”.(US10)

Consequently, several respondents acknowledged the need to provide reassurance to the family due to the perceived impact of the uncertain outcome for the family.

*“…empathize that this is a not a good place for the family*”.(Europe52)

Many respondents also indicated that once the CRMS/CFSPID designation has been determined, they would inform the child’s family that the child is well and would be unlikely to require treatment but would need to be followed up to monitor any changes on their health status that may be indicative of them converting to a CF diagnosis or developing a CFTR related disorder.

“*Their child is likely to remain well, should not need treatment, and will be unlikely to develop symptoms suggestive of CF but this may change over time and this means that their child will need review through their childhood*”.(Europe41)

Despite emphasizing that the child was well and did not have CF, respondents, particularly in the US, reported that they would discuss signs and symptoms of CF with the family during this initial consultation.

“*They have a child… that is indeterminate for full diagnosis of CF but it may develop over time, therefore necessitating we follow them intermittently to monitor for disease before obvious signs and symptoms, specifically end organ injury…*” (US18)

Most respondents indicated details of children who had been given a designation of CMRS/CFSPID would be stored on the relevant CF Registry (*n* = 75, 74%), *n* = 15 (15%) stated there was not a national database where the child’s details could be stored, and *n* = 4 (4%) stated these would be stored on a separate CRMS/CFSPID registry, *n* = 7 (7%) did not respond.

### 3.3. Ongoing Management of Children with a CRMS/CFSPID Designation

Clinicians were asked a series of closed-ended questions about the ongoing management of children with a CRMS/CFSPID designation. Responses to these are summarized in Table 3. The Chi-Square test was used to determine any statistically significant differences between management strategies in the US and Europe (Canada was excluded from analysis due to the low response rate). Significantly more respondents in the US compared to Europe reported they would not manage children with an intermediate sweat chloride value (30–59 mmol/L) and one or no CFTR mutations, differently when compared with those children with a normal sweat chloride (<30 mmol/L) and two CFTR mutations, at least one of which has unclear phenotypic consequences (χ^2^ = 14.631, d.f. 1, *p* < 0.01). For those who would manage them differently, responses to an open-ended question revealed this was reportedly due to those with an intermediate sweat chloride being considered more likely to display symptoms and convert to a CF diagnosis or develop a CFT- related disorder. Respondents stated their intention would therefore be to follow up these infants more frequently, although the intended frequency was variable. Other differences included offering sodium supplementation, particularly in the summer, oral antibiotics for a new cough, regular respiratory cultures, and advice regarding ‘high risk’ activities such as using aerated baths.

“*… we would advise oral antibiotics for a new cough lasting 48 h, and would take a cough swab if lasting 2 weeks….to avoid activities at high risk for CF pathogens (eg jacuzzi)…we advise there is a small chance (possibly around 10%) of them at some point in the future being recategorized as atypical CF*”.(Europe14)

Almost all (*n* = 92, 91%) respondents reported that they would follow up infants with a CRMS/CFSPID designation in a specialist CF clinic. Reasons for not seeing these infants in a specialist CF clinic included: viewing it as unnecessary (*n* = 3, 3%), children being seen by a specialist but not in a CF clinic (*n* = 3, 3%), and concern this may confuse parents and make them think their child had CF (*n* = 2, 2%), *n* = 1 (1%) did not provide a reason). Of those who did see children in a specialist CF clinic, most (*n* = 85, 92%), reported policies were in place to ensure infants with a CRMS/CFSPID designation were not exposed to an increased risk of cross-infection while attending clinic appointments (*n* = 6, 7%) stated there would not be specific infection control policies and *n* = 1, (1%) did not respond to this question. Reasons for not having policies in place included: lack of capacity (time, space, or staff) (*n* = 3, 3%), and not feeling it is necessary due to there being no evidence that infants with a CRMS/CFSPID designation are at increased risk of infection (*n* = 3, 3%). Where policies did reportedly exist, these were multifaceted and most commonly consisted of staff washing their hands before and after the consultation (*n* = 84, 99%), each child is placed in a separate room (*n* = 69, 81%), the staff is required to wear apron and gloves during the consultation (*n* = 51, 60%) as well as the child is seen at the beginning or end of the CF clinic (*n* = 29, 34%) or in a separate clinic to children with CF (*n* = 19, 22%).

Almost all respondents (*n* = 97, 96%) reported offering children with a CRMS/CFSPID designation a repeat sweat test following the initial consultation (*n* = 2, 2% stated they did not and *n* = 2, 2% did not respond). Of those who reported offering a repeat sweat test, 48 (49%) stated this would happen when the child was six months of age, and a further 22 (23%) stated this would happen when the child was 12 months of age. The remaining respondents (*n* = 27, 28%) undertook more than one repeat sweat test on these infants, the frequency of which ranged from every six months until the child reached seven years of age to this being variable and/or as needed.

In terms of information gathering prior to review of children with a CRMS/CFSPID designation, *n* = 86 (85%) of respondents reported they would consult the CFTR-2/CFTR-France website prior to the review, *n* = 8, (8%) stated they would not, and *n* = 7 (7%) did not answer. Of those who would not access the CFTR-2/CFTR-France website *n* = 3 (38%) stated this was because it was viewed as being too difficult or time-consuming to access, *n* = 1 (13%) stated they viewed it on an as needed basis, rather than prior to each review and *n* = 4 (50%) did not provide a reason. For those who reported that they would check the CFTR-2/CFTR-France website prior to the review, *n* = 75 (87%) provided specific reasons for doing so which included: obtaining up to date information about specific mutations (*n* = 36, 41%), informing clinical decision making/management of the child (*n* = 22, 26%), gaining information about the prognostic outcomes associated with different mutations (*n* = 12, 14%), and to facilitate providing up to date information to the family (*n* = 5, 6%).

Reported timing and frequency of reviews was variable (ranging from three-six-monthly to not until the child reached age five-six years of age) but for those infants with no clinical concerns, most respondents (*n* = 80, 79%) indicated the intention to review them annually. Respondents indicated they took a multitude of factors into consideration when determining how frequently they would undertake reviews of children with a CRMS/CFSPID designation. These included: clinical assessment (including respiratory, abdominal and nutritional assessment) (*n* = 94, 93%), the sweat chloride value (*n* = 81, 80%), parental anxiety (*n* = 80, 79%), respiratory cultures (*n* = 66, 65%), the NBS result (*n* = 53, 52%), pulmonary function (*n* = 36, 36%), chest x-ray findings (*n* = 30, 30%), chest computerized tomography scan results (*n* = 8, 8%), genotype (*n* = 5, 5%), local guidelines (*n* = 2, 2%) and pancreatic elastase (*n* = 1, 1%). The most frequent tests or measurements that would reportedly be undertaken as part of or in preparation for, review appointments included: respiratory cultures (*n* = 85, 84%), pulmonary function tests (*n* = 81, 80%)—most commonly these were done once the child reached >5 years of age and fecal elastase (*n* = 65, 64%). Of those who would obtain respiratory cultures, *n* = 37 (43%) stated they would perform these annually, *n* = 15 (18%) stated they would perform these at every visit and *n* = 13 (15%) stated they would only perform these if the child was symptomatic (for the remainder, *n* = 20, 24%, timing of respiratory cultures was variable).

For children with a CRMS/CFSPID designation who reach six years of age in good health with normal growth, lung function and imaging, and normal sweat chloride values, and are therefore unlikely to convert to a diagnosis of CF, *n* = 42 (41%) reported they would continue regular specialist review either as part of the CF clinic or in a separate clinic (this could be ‘virtually’ for example as an annual telephone call or video consultation). A further *n* = 22 (22%) reported they would discharge the child from CF specialist care, but offer a further isolated specialist review as the child reaches adolescence (at the age of around 14–16 years and *n* = 21 (21%) reported they would discharge the child from CF specialist care, with follow-up in primary care by a PCP/GP. For *n* = 9, (9%), responses varied and indicated no consistent policy existed, *n* = 7 (7%) did not respond.

The majority (*n* = 88, 87%) of respondents agreed that for children with CRMS/CFSPID who are discharged from specialist care, a subsequent review should be considered when the child is a young adult, to communicate the information directly to them, *n* = 5 (5%) felt they should not be offered a review and *n* = 8 (8%) did not respond. For those who felt children should be reviewed, nine (10%) felt the review should take place when the child was aged between 6–12 years, the majority *n* = 68, (77%) felt this should happen between the ages of 13–18 years with *n* = 19 (28%) of these believing it should happen when the child reached age 18 years, and *n* = 6 (7%) felt the review should happen between the ages of 18–21 years, *n* = 5 (6% did not respond).

### 3.4. Support Outside the CF Team

Responses indicated that advice regarding when parents should seek medical advice about their child and advice given to the child’s PCP/GP were consistent; this is summarized in Table 4. In terms of health promotion advice, *n* = 87 (86%) respondents stated they would advise parents that their child should follow the national immunization programme (*n* = 7, 7% did not respond). In addition, *n* = 86 (85%) stated they would advise parents that their child should not be exposed to cigarette smoke, *n* = 83 (82%) stated they would advise children and their families to adopt a healthy lifestyle consistent with national guidance on exercise, nutrition and other aspects of public health policy, *n* = 1 (1%) stated they would promote breastfeeding over the age of six months and advise parents to avoid community care for their child for the first two years of their life, *n* = 1 (1%) stated that they would advise families that their environment should be pseudomonas free and *n* = 1 (1%) said they would provide advice regarding family planning (*n* = 9, 9% did not respond).

Significantly more respondents in Europe compared to the US reported they would offer families a referral for genetic counselling (χ^2^ = 5.792, d.f. 1, *p* = 0.02). Of the 85 (84%) respondents who said they would offer a referral, *n* = 63 (74%) stated they would discuss this at the initial consultation with the family, *n* = 10 (12%) stated they would discuss this during the annual review, *n* = 5 (6%) said this would be dependent on the family, *n* = 4 (5%) said they would do this during the first year and *n* = 3 (4%) stated it would be during their first visit. In terms of how long it would take for the family to be seen following the referral, *n* = 13 (15%) reported that the family would be seen by the genetic service during the initial visit following the positive NBS result, *n* = 64 (75%) reported the family would be seen before the baby reached 6 months of age, and *n* = 4 (5%) reported they would be seen when the baby was between 6–12 months of age. The remaining *n* = 4 (5%) respondents indicated this would vary.

## 4. Discussion

Identification of infants with an inconclusive diagnosis after a positive NBS result, designated CRMS/CFSPID leads to uncertainty for both families and healthcare professionals [6]. Recent, updated guidance on the management of these infants aimed to ensure more consistent and appropriate care pathways are employed [13]. Results of the present study indicated that while respondents reported minor deviations in practice, they were cognizant of recommendations in the updated guidance [13], and for the most part, the intention was to implement these into practice consistently internationally. However, this is not consistent with studies that have collected clinical data (rather than reported practice) for children with a CRMS/CFSPID designation which have demonstrated inconsistent practice both in relation to sweat testing and follow-up [14,16].

Respondents in the present study reported using a multitude of tests during the initial assessment to confirm the CRMS/CFSPID designation including clinical evaluation, sweat testing, extended CFTR analysis if the genotype was incomplete, and a collection of stool sample for measurement of fecal elastase; these were commensurate with those recommended in the update guidance [13]. Following the initial assessment and due to the uncertainty associated with the CRMS/CFSPID designation, it is vital that the initial communication of the CRMS/CFSPID result to the family is clear and consistent [13]. This is important since previous research has highlighted that poor communication of positive NBS results to families can influence parental outcomes in the short term [21,22,23,24,25,26] but may also have a longer-term impact on children and families [27]. In the present study, as per the updated guidance [13], respondents reported that they would emphasize to parents that their child is well, does not have CF but will need to be followed up. Respondents also acknowledged the uncertainty the CRMS/CFSPID designation created for families and were empathetic with regard to their information needs in relation to this.

In the present study, significantly more respondents in the US compared to Europe (χ^2^ = 14.631, d.f. 1, *p* < 0.01) reported they would not manage children with a normal sweat chloride (<30 mmol/L) compared to children with an intermediate sweat chloride value (30–59 mmol/L), differently. Those who reported they would be managed differently indicated this was due to those with an intermediate sweat chloride being considered more likely to display symptoms and convert to a CF diagnosis or develop a CFTR related disorder. This reflects evidence which suggests infants with an initial intermediate sweat chloride concentration are more likely to convert to a CF diagnosis than those in whom the initial value was normal [8,9,12].

Almost all (*n* = 92, 91%) respondents in the present study reported they would follow up infants with a CRMS/CFSPID designation in a specialist CF clinic which would enable them to follow recommendations regarding prevention of potential cross-infection [13]. This contradicts findings from a study in Switzerland which found that in practice, only half of the children with a CRMS/CFSPID designation were receiving ongoing follow-up by a PCP/GP or pediatrician [16]. In terms of ongoing clinical management, the updated guidance recommends that children should have repeat sweat testing performed when the child is aged six months, two years, and six years of age; respiratory, abdominal, and nutritional assessment when the child is 6 and 12 months of age and then annually and respiratory culture and chest imaging if clinically indicated [13]. In the present study, most participants reported that the repeat sweat test would take place when the child was either six months (*n* = 48, 49%) or 12 months of age (*n* = 22, 23%) with only *n* = 27 (28%) reporting that more than one repeat sweat test would be undertaken; the frequency of which ranged from every six months until the child reached seven years of age to this being variable and/or as needed. This is contracted in a study reporting actual practice in Switzerland which found that nearly half of the children with a CRMS/CFSPID designation had no follow up with a second sweat test [16]. The findings of the present study indicated that the frequency with which reviews were undertaken would be influenced by parental anxiety. Few studies have explored parental experiences and responses to being told their child has a CRMS/CFSPID designation. Those that have, have highlighted that the uncertainty associated with receiving a CRMS/CFSPID designation for their child led to emotional distress. For instance, a study in America found that uncertainty associated with ambiguity as the screening and diagnostic results were perceived to be contradictory, the unknown disease trajectory and difficulty distinguishing between normal childhood problems from those associated with CF was central to parent’s experiences of receiving their child’s CRMS/CFSPID designation [17]. Similarly, parents in a more recent study conducted in England described communication of the CRMS/CFSPID result as intrusive and traumatic followed by feelings of fear and grief [18]. Therefore, acknowledging the potential emotional distress caused by this uncertain outcome and using this to inform timing and frequency of reviews was considered a positive outcome of this work.

Current evidence suggests that for most children who are six years of age in good health with normal growth, lung function and imaging, and normal sweat chloride values (<30 mmol/L), conversion to a diagnosis of CF is unlikely [12]. Despite this, nearly half of respondents *n* = 42 (41%) in the present study reported they would continue regular specialist review either as part of the CF clinic or in a separate clinic. Furthermore, the majority (*n* = 88, 87%) of respondents agreed that children with CRMS/CFSPID should be offered a further review when the child is a young adult, to enable information about the CRMS/CFSPID designation to be communicated directly to them. While this reflects the update guidance in terms of a review taking place [13], the age at which this review should take place was variable and ranged from 6–21 years.

In terms of care outside of the immediate CF team provision, the updated guidance [13] suggests a referral for genetic counselling should take place at the initial assessment and this was reflected in responses in the present study. Interestingly, significantly more respondents in Europe compared to the US stated they would offer families a referral for genetic counselling (χ^2^ = 5.792, d.f. 1, *p* = 0.02); the reasoning behind this was not explored. Respondents also reported that information given to parents regarding when to seek additional medical advice for their child was very similar to advice given to PCPs/GPs. This is important since it has been shown that providing high-quality information and reducing perceived power imbalances between health professionals and parents/children can facilitate shared decision-making in pediatric practice [28]. However, little attention was given to the risk of pancreatitis to either parents or PCPs/GPs despite this being one of the conditions most well characterized as a CFTR-related disorder [29,30].

Overall, the results of the present study indicate that clinicians are cognizant of the updated guidance [13] and are keen to ensure these are being consistently implemented in practice with minimal variation. However, this does not reflect the findings of studies that have explored actual practice [14,16]. Many reasons could account for this disparity between actual and reported practice. The present study targeted a purposeful sample of clinicians with a specific interest in CF NBS internationally who are more likely to be aware of the updated guidance and the importance of consistent implementation in practice [13]; different results may have been obtained if all clinicians involved in managing children with a CRMS/CFSPID designation were surveyed internationally. In addition, the sample were very experienced in looking after children with CF; the majority of respondents were doctors (*n* = 76, 75%), and most had worked with children with CF for over 15 years (*n* = 73, 72%). Again, different results may have been obtained from a more varied sample with less clinical experience. Respondents in the present study were clearly familiar with the recommendations contained within the updated guidance [13]; it is known that in such instances, respondents are more likely to answer in a way that would be viewed favorably by others [31]. Finally, the present study presents reported rather than actual practice, and as such, there is no evidence to support or refute statements made by respondents.

In summary, it is reassuring that workers in the field have acknowledged and appreciate the new guidance for the management of infants with a CRMS/CFSPID designation, but further research is needed to determine if these findings are reflective of both a wider audience of clinicians and actual (rather than reported) practice to ensure the guidance is being implemented in practice consistently and as intended.

## Figures and Tables

**Table 1 IJNS-08-00005-t001:** Job title of study participants.

Job Title	US	Canada	Europe
Centre Director	6		3
Doctor	30	3	43
Laboratory Staff	1		
Newborn screening co-ordinator			1
Nurse/Nurse practitioner		3	1
Paediatric Programme Director	1		
Professor/Associate Professor	1		7
Research Scientist			1
Total	39	6	56

**Table 2 IJNS-08-00005-t002:** Years working with children with CF and CRMS/CFSPID.

Number of Years	Working with Children with CF			Working with Children with CRMS/CFSPID		
	US	Canada	Europe	US	Canada	Europe
0–4		1	2	2	1	11
5–9	5	1	7	9	1	11
10–14	6		6	22	3	15
15–19	6	1	11	2	1	14
20–24	6	3	11			3
25–29	6		7			1
30–34	8		8			1
35–39	1		4			
40–44	1			1		
Not answered				3		
Total	39	6	56	39	6	56

**Table 3 IJNS-08-00005-t003:** Summary of current management for children with CRMS/CFSPID designation (* *p* < 0.05, ** *p* < 0.01).

Question	Responses: US *n* = 39, Canada *n* = 6, Europe *n* = 56									Total
	Yes			No			Unanswered			
	US	Canada	Europe	US	Canada	Europe	US	Canada	Europe	101
Do you manage children with (a) Normal sweat chloride (<30 mmol/L) and two CFTR mutations, at least one of which has unclear phenotypic consequences (b) Intermediate sweat chloride value (30–59 mmol/L) and one or no CFTR mutations, differently	5 **	2	27 **	33 **	3	29 **	1	1	0	101
Do you follow up infants with a CRMS/CFSPID designation in a specialist CF clinic?	37	5	50	2	1	6	0	0	0	101
Do any policies exist to ensure the infant is not exposed to any increased risk of cross infection?	36	4	45	1	1	4	2	1	7	101
Do you offer these infants a repeat sweat test at any point?	39	6	52	0	0	2	0	0	2	101
Do you review the CFTR-2/CFTR-France website prior to the review?	32	4	50	4	0	4	3	2	2	101
Do you do any respiratory cultures at the review appointment or at any other times?	35	5	48	1	0	6	3	1	2	101
Do you offer families a referral for genetic counselling?	29 *	5	51 *	7 *	0	2 *	3	1	3	101
Is there a national database where the infants’ details can be stored?	35	5	39	1	0	14	3	1	3	101
Do you think a review for children with CRMS/CFSPID who are discharged from specialist care, should be organised when the child is a young adult to communicate information directly to them, as per recent guidance?	33	5	50	2	0	3	4	1	3	101

**Table 4 IJNS-08-00005-t004:** Advice for parents and primary care practitioners (PCPs)/general practitioners (GPs) regarding when parents should seek medical advice.

Symptoms	Advice to Parents *n*, (%)	Advice to PCPs /GPs *n*, %
Persistent respiratory symptoms lasting more than 2 weeks	94 (93)	86 (85)
Failure to gain weight	90 (89)	86 (85)
Persistent loose stools	80 (79)	80 (79)
Sinus issues	3 (3)	4 (4)
Any other concerns	3 (3)	4 (4)
Pancreatitis	2 (2)	3 (3)
Digestive symptoms	2 (2)	1 (1)
Evidence of salt loss	1 (1)	2 (2)
Jaundice	1 (1)	1 (1)
Abdominal pain		
Constipation		1 (1)
Results of swabs		1 (1)

## Data Availability

The data presented in this study are available on request from the corresponding author. The data are not publicly available due to ethical constraints.

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
