# Peer review of "International Approaches to Management of CFTR-Related Metabolic Syndrome/Cystic Fibrosis Screen Positive, Inconclusive Diagnosis"

_2409-515X, 2022, doi:10.3390/ijns8010005_

Round 1

Reviewer 1 Report

This is a thoughtful, comprehensive and well-written analysis of health professional's  approaches to a difficult and increasingly common clinical scenario, and how well these correspond to recent guidelines. The authors have acknowledged the likely difference between reported and actual actions as well as selection bias amongst what appears to be a very experienced group of clinicians who chose to complete the questionnaire. 

My suggestions are minor

  1. The proportion of respondents per region is not immediately obvious in Table 4, the authors may consider revising this so that eg. "5" US row 1 is expressed as a % of "US yes respondents" or 5/39.
  2. Details of the questionnaire could be added as an Appendix

Author Response

Point 1: The proportion of respondents per region is not immediately obvious in Table 4, the authors may consider revising this so that eg. "5" US row 1 is expressed as a % of "US yes respondents" or 5/39.

RESPONSE: The total number of responses per region has been added at the top of the table as a reminder to avoid the table becoming too 'busy'

Point 2: Details of the questionnaire could be added as an Appendix

RESPONSE: The questionnaire has been provided 

Reviewer 2 Report

Recently an updated guidance for the management of children with CRMS/CFSPID was published by the ECFS experts. The reviewing paper is an interesting investigation using the online questionnaire distributed via Qualtrics Survey Software to explore CF health professionals’ reported experiences and approaches to managing children who receive a designation of CRMS/CFSPID following a positive NBS result.  The results of the present study indicate that clinicians are cognizant of the updated guidance and are keen to ensure these are being consistently implemented in practice with minimal variation. The study is reassuring that workers in the field of CF have acknowledged and appreciate the new guidance for the management of infants with a CRMS/CFSPID designation, but further research is needed to determine if these findings are reflective of both a wider audience of clinicians and actual (rather than reported) practice to ensure the guidance is being implemented in practice consistently and as intended.

I have one main question and will mention several minor inaccuracies and misspellings in the text that need to be corrected. 

  1. The clinicians were divided into 3 groups: from the United States - 39; from Canada - 6; and from Europe - 56. What is the reason for keeping a separate group for Canada? Can't these 6 clinicians be added to the US group? For example, the UK is included in the European group. If Canadians are with the US (Notrth American Group) there is no need to exclude them from the analysis in some places (for example, lines 169-170).  
  2. In ceveral places in the text a normal chloride value is mentioned as (30 mmol/l)  and it should be (< 30 mmol/l): on lines - 40, 173, in Table 4, 333. Please, check all the text.
  3. Please, check the text on lines 159-163. I think there are missing brackets after 15% and 4%?
  4. Line 167 - Reponses... (responses?)
  5. Line 169 - ... form (from?) analysis    
  6. Line 171 - ... a (a) an intermediate 
  7. Lines 355, 361 and 362 - a CFSPID (a CRMS/CFSPID?)
  8. Line 367 - ...for more children who six years ... (... who are six...?)

Author Response

Thank you for your positive feedback. 

Point 1: The clinicians were divided into 3 groups: from the United States - 39; from Canada - 6; and from Europe - 56. What is the reason for keeping a separate group for Canada? Can't these 6 clinicians be added to the US group? 

RESPONSE: There were two main reasons for separating the Canadian responses from the US. The first was down to the way in which the survey was administered and therefore the way respondents were recruited. In the US, the survey link was sent by the US CF Foundation to all US CF Center Directors. Respondents from Canada were recruited via the ECFS NSWG. The second is due to the different healthcare systems in the US and Canada; it was felt inappropriate to combine responses given that this may influence the management of these children. 

Point 2: In several places in the text a normal chloride value is mentioned as (30 mmol/l)  and it should be (< 30 mmol/l): on lines - 40, 173, in Table 4, 333. Please, check all the text.

RESPONSE: Thank you for noticing this, this has been checked and corrected throughout. 

Points 3-8: Please, check the text on lines 159-163. I think there are missing brackets after 15% and 4%? Line 167 - Reponses... (responses?). Line 169 - ... form (from?) analysis. Line 171 - ... a (a) an intermediate. Lines 355, 361 and 362 - a CFSPID (a CRMS/CFSPID?). Line 367 - ...for more children who six years ... (... who are six...?)

RESPONSE: Thank you for alerting us to these, these have all now been corrected. 

Reviewer 3 Report

I read this paper with great interest. The authors report the results from a questionnaire completed by both European and American CF clinicians. There is a tendency to follow the most recent recommendations on CRMS / CFSPID (Barben et al. JCF 2021) albeit with some exceptions and differences between Europeans and Americans (this is one of the most interesting data of the work, as well reported in table 4). Obviously a bias of this paper is that many clinicians are experienced in CF and therefore well documented, but this has already been included among the limitations of the paper.
My opinion is positive, I think this is a useful paper and this Journal is ideal for it.
Here are some of my advices to improve the work and make it easier to read:
- introduction: well done, complete. I have nothing to add.
- material and methods: short, but table 4 helps because the questions are directly reported
- results: in my opinion they are too long and difficult to read until the end. I would delete tables 1 and 2 and simplify the results by adding some summary tables. The results show that 28% of the respondents would take the sweat test more frequently (I agree with this point). I believe that a comment in discussion is necessary and useful. Finally, perhaps little attention emerges on the risk of pancreatitis in CRMS / CFSPID (see table 5).

- tables: with regard to questions 1 and 7 (table 4) it seems the opposite of what was reported in the results and discussion, ie that the American respondents do NOT differentiate management based on the sweat test or less frequently they provide genetic counseling. Maybe I could be wrong, I ask for a clarification.
- discussion: it is well based on the results of the paper.

Author Response

Thank you for your positive feedback. 

Results:

Point 1: In my opinion they are too long and difficult to read until the end. I would delete tables 1 and 2 and simplify the results by adding some summary tables.

RESPONSE: We would like to keep Table 1 as we feel it provides important demographic information about the sample.  However, we have combined tables 2 and 3 to illustrate the difference between respondents' experience with CF and CRMS/CFSPID. It is difficult to tabulate many of the other results as to make sense, they require accompanying explanations as provided in the text. 

Point 2: The results show that 28% of the respondents would take the sweat test more frequently (I agree with this point). I believe that a comment in discussion is necessary and useful.

RESPONSE: This has been added to the discussion (lines 352-357)

Point 3: Finally, perhaps little attention emerges on the risk of pancreatitis in CRMS / CFSPID (see table 5).

RESPONSE: This has been added to the discussion (lines 394-396) with accompanying references. 

Tables:

Point 1: with regard to questions 1 and 7 (table 4) it seems the opposite of what was reported in the results and discussion, ie that the American respondents do NOT differentiate management based on the sweat test or less frequently they provide genetic counselling. Maybe I could be wrong, I ask for a clarification.

RESPONSE: Thank you for noticing this, it has not been corrected in the results (172-175 and 294-295) and discussion (lines 333-335 and 387-389).